# Laser Ablation-Assisted Synthesis of Poly (Vinylidene Fluoride)/Au Nanocomposites: Crystalline Phase and Micromechanical Finite Element Analysis

**DOI:** 10.3390/polym12112630

**Published:** 2020-11-09

**Authors:** Yasin Orooji, Babak Jaleh, Fatemeh Homayouni, Parisa Fakhri, Mohammad Kashfi, Mohammad Javad Torkamany, Ali Akbar Yousefi

**Affiliations:** 1College of Materials Science and Engineering, Nanjing Forestry University, Nanjing 210037, China; yasin@njfu.edu.cn; 2Department of Physics, Bu-Ali Sina University, Hamedan 65174, Iran; f.bazak@gmail.com; 3Instrumentation Research Group, Niroo Research Institute (NRI), Tehran 1468613113, Iran; 4Mechanical Engineering Department, Ayatollah Boroujerdi University, Boroujerd 69199-69737, Iran; mkashfi12@gmail.com; 5Iranian National Center for Laser Science and Technology (INLC), Tehran 14665-576, Iran; mjtorkamany@yahoo.com; 6Plastic Materials Department, Faculty of Polymer Processing, Iran Polymer and Petrochemical Institute, P.O. Box 14965/115, Tehran 13115-14977, Iran; a.yousefi@ippi.ac.ir

**Keywords:** Au nanoparticles, micromechanical model, piezoelectric, PVDF

## Abstract

In this research, piezoelectric polymer nanocomposite films were produced through solution mixing of laser-synthesized Au nanoparticles in poly (vinylidene fluoride) (PVDF) matrix. Synthetization of Au nanoparticles was carried out by laser ablation in *N*-methyle-2-pyrrolidene (NMP), and then it was added to PVDF: NMP solution with three different concentrations. Fourier transformed infrared spectroscopy (FTIR) and X-ray diffraction (XRD) were carried out in order to study the crystalline structure of the nanocomposite films. Results revealed that a remakable change in crystalline polymorph of PVDF has occurred by embedding Au nanoparticles into the polymer matrix. The polar phase fraction was greatly improved by increasing the loading content of Au nanoparticle. Thermogravimetric analysis (TGA) showed that the nanocomposite films are more resistant to high temperature and thermal degradation. An increment in dielectric constant was noticed by increasing the concentration of Au nanoparticles through capacitance, inductance, and resistance (LCR) measurement. Moreover, the mechanical properties of nanocomposites were numerically anticipated by a finite element based micromechanical model. The results reveal an enhancement in both tensile and shear moduli.

## 1. Introduction

Additive manufacturing processes and modeling of polymers, alloys, and compounds have profoundly influenced both academy and industries [1,2,3,4]. Poly (vinylidene fluoride) (PVDF) is a famous semi-crystalline polymer that has attracted much attention because of its almost unique physical and chemical properties. The electroactive properties of PVDF together with its high chemical resistance, high mechanical strength, ease of processing, and low cost [5,6,7,8,9,10,11] have made it an attractive, smart polymer which can be used in nanogenerators for sensors [12] and energy harvesting [13,14] applications.

Piezoelectric properties of PVDF are dependent on its crystalline structure. This polymer has three common crystalline phases; namely, non-polar *α* phase, polar β phase and polar γ phase, which are different in macromolecular chain conformations [15]. Since the second and the third phases are responsible for piezoelectric properties, they are PVDF’s most favorable phases. β and γ phases cannot be shaped naturally, while *α* phase-dominated PVDF film is easily obtained because of its thermodynamic stability [16]. The crystalline *α* phase can be transformed to polar phases using several methods such as electric poling [17], mechanical stretching [18], electrospinning [19,20,21], and nanofiller addition [22]. As described in the literature, nanofiller addition has been introduced as an interesting approach without limitation in other methods such as not being cost-effective, particularly at large scale production. Adding various nanofillers such as Fe_3_O_4_ [23], TiO_2_ [24], Al [25], and diamond nanoparticles [16] in order to promote the polar phases of the PVDF polymer films has been recently reported.

On the other hand, due to its great potential application in MEMS and electronic devices, improvement of the dielectric properties of PVDF has also attracted a lot of attention [26], and the best way to improve its dielectric properties, while maintaining its mechanical properties, is adding conductive nanofillers to the polymer matrix [27].

For practical application, it is vital to use a conductive nanofiller that improves the piezoelectric and dielectric properties. In some researches, Conductive Au nanoparticles have been used as filler in PVDF polymer matrix, as shown in Table 1.

The research works including [28,29,32,33,34,35,36] stated the polymorphism change in PVDF by adding Au nanoparticles, whereas the phase content has not been reported; therefore, the amount of change in β phase is not quantitatively known. The electroactive property of PVDF is to be studied with more details and consideration since it is its most important property. Reference [30] concentrated on electromagnetic properties of PVDF by adding Au nanoparticles, and piezoelectric and dielectric properties have not been taken into consideration. In addition, in all the afore-mentioned works, except reference [33], chemical techniques have been utilized to synthesize Au nanoparticles. According to the literature [37,38], in comparison to common chemical techniques, laser ablation in liquids has been introduced as a promising technique to synthesize metal colloids. The absence of chemical reagents is the most remarkable advantage of this technique, and another advantage is the well-dispersion of Laser-ablated nanoparticles in the liquid phase [37,38].

As mentioned before, since these nanocomposites are to be applied in nanogenerators for sensor and energy harvesting applications, it is important to study their mechanical properties that have not been reported for PVDF/Au nanocomposites yet. The micromechanical analysis based on the finite element method (FEM) for a representative volume element (RVE) is an attractive tool for the prediction of the mechanical properties of composite materials. The micromechanical analysis not only reduces the money-consuming experiments and measurement devices such as digital image correlation (DIC) system for determining the Poisson’s ratio, but also saves much time and many sources. This method can accurately predict all mechanical properties for any weight content of the Au nanoparticle phase by performing each validated simulation, and it is rarely used for other PVDF-based nanocomposite films [39,40].

Here, PVDF nanocomposites filled with Au nanoparticles at low concentrations were prepared using the solution casting method. Au nanoparticles were synthesized by laser ablation in NMP, and their structure, morphology and *β*-phase content were experimentally studied by SEM, FTIR and XRD. Thermal properties were investigated by TGA analysis, and dielectric properties were measured by LCR meter. The mechanical properties of the composite were then predicted by using the micromechanical scheme by considering a RVE solved by finite element analysis.

## 2. Materials and Methods

### 2.1. Materials

PVDF (MW = 534,000 g/mol) was supplied by Sigma Aldrich (Lyon, France). Dimethylformamide (DMF) and *N*-methyl-2-pyrrolidine (NMP) purchased from Merck (Hohenbrunn, Germany) were used as the solvents. Au nanoparticles were synthesized through the following procedure:

### 2.2. Synthesis of Au Nanoparticles

Au nanoparticles were synthesized by laser ablation of a gold metal plate (99.99%) in NMP without adding any chemical additives. The nanoparticles were placed at the bottom of a glass vessel which contained 3 mL NMP. A high-frequency Nd:YAG laser (1064 nm) with 5 mJ pulse energy and 240 ns pulse duration (FWHM) at a repetition rate of 2 kHz was used in order to conduct the laser ablation process. By irradiation of the laser beam, gradually, the color of the liquid turned to purple because of the formation of gold nanoparticles. The color change occurs due to surface plasmon resonance redshift of Au nanoparticles.

### 2.3. Fabrication of the Nanocomposite Films

PVDF solution was prepared by mixing its powder in DMF of a mass ratio of 10/90 and by stirring the solution for 3 h at room temperature until PVDF was fully dissolved. DMF was chosen because PVDF is dissolved well in DMF; however, the ablation rate of the gold target in the DMF is low. Since the boiling point of DMF (153 °C) is lower than that of NMP (202 °C), NMP was chosen to generate Au nanoparticles by laser ablation. To disperse Au nanoparticles inside the polymer matrix homogeneously, the nanoparticle colloid was first put in an ultrasound bath for almost 1 h. Then, the amount considered for the PVDF/DMF solution was added to nanoparticles colloid, and the obtained solution was stirred to make sure the polymer and Au nanoparticles have been completely mixed. The mass ratio of Au nanoparticles to PVDF was varied as 0.05%, 0.1% to 0.5%. In the next step, the prepared mixture was transferred onto a petri dish and it was placed in an oven for about 6 h at 120 °C in order to complete the crystallization of the nanocomposite and to remove DMF and NMP. After drying the solution, a comparatively uniform bright purple thin film is obtained. The nanocomposite films were labeled as PVDF/*x*%Au, where *x* illustrates the weight fraction of Au nanoparticles, and the net PVDF films are also labeled as PVDF.

### 2.4. Material Characterization

FTIR spectroscopy (Perkin Elmer, SPECTRUM-GX, USA, Mid infrared Source: wire coil; Mid infrared Detector: FR-DTGS with KBr window) was employed in order to determine the crystalline phase of nanocomposite films in the wavenumber range of 4000 to 400 cm^−1^ at a resolution of 4 cm^−1^. X-ray observations were carried out by a Philips powder diffractometer type PW 1373 (Philips, Amsterdam, The Netherlands) with a graphite mono-chromator crystal. The diffraction patterns were collected in the 2θ range of 10–60 degree with a scanning speed of 2 degree/min and the X-ray wavelength of 1.5405 Å. The morphology of the films was studied by FESEM (Field Emission Scanning Electron Microscopy, TESCAN MIRA3-XMU, Brno-Kohoutovice, Czech Republic). Thermal analysis was conducted by a Perkin Elmer TG-DTA system in a temperature range of 25 to 600 °C under nitrogen atmosphere (70 cm^3^·min^−1^) with the rate of 10 °C/min. LRC meter Agilent 4258 A (Hewlett Packard, Palo Alto, CA, USA) measured the dielectric constants of nanocomposite film, and a frequency range of 75 kHz to 2 MHz was used.

### 2.5. Finite Element Simulation

FEM is an effective tool to simulate the mechanical response and properties of structures [41]. Recently, FEM has been employed to predict the mechanical properties of composite materials by applying simple loading conditions on a reprehensive volume element (RVE) even for the behavior of anisotropic materials [42]. In the present study, the mechanical properties of fabricated nanocomposite are predicted by using the micromechanical scheme by considering an RVE which is solved by a FEM. In this method, the RVE model is constructed with two distinct phases called nanoparticle and matrix, which are shown in Figure 1. Periodic boundary conditions are considered the model, and a ramp displacement load is applied to a remote node in an attempt to simulate the uniaxial condition.

Updated Lagrangian four-noded tetrahedral elements with linear interpolation functions isoparametric element (Element type 18 in Marc) were considered for simulation by enabling large strain deformation option. These types of elements are normally used to model solid objects. The stiffness matrix is integrated by using a single integration point at the centroid. The distributed load on a face is integrated by using a single integration point at the centroid of the face. Three global degrees of freedom *u*, *v*, and *w* per node are defined for this type of element along *x*, *y* and *z* directions. The elements of the constructed model are sufficiently refined so that a desirable convergence is obtained on the determined results. To increase the mesh accuracy, three-time refinement steps were conducted so that a uniform mesh refinement process was obtained by using automatic mesh generators via Mentat preprocessor. Internal mesh coursing is also activated in order to reduce the computational cost. After mesh dependency analysis, the optimum number of elements and nodes are achieved 30,380 (348 for inclusions and 30,032 for the matrix) and 45,481 (136,443 degrees of freedom), respectively. The mesh quality is evaluated by *ρ* parameter, which is defined by the ratio between the length of the shortest and longest edges of elements. The maximum and mean values of *ρ* parameter are obtained 0.97 and 0.64, respectively. Figure 2 shows the *ρ* parameter distribution for the elements constructed in the considered RVE. As the figure suggests, a normal distribution is clearly observed. In order to solve the FE model, Newton-Raphson method was selected to solve the nonlinear equilibrium equations using the implicit algorithm implemented in MSC Marc. It takes about 20 min to solve each simulation with an Intel i7-6700K CPU with 16 GB of RAM.

In order to validate the constructed FE model, a micromechanical-based RVE is considered to investigate the results reported by Srivastava, Maiti [43]. As previously presented, Table 2 shows the 3D model and constructed elements of the micromechanical model of CaCu_3_Ti_4_O_12_ (CCTO) nanoparticle inclusion in the PVDF matrix. The particles are assumed in the same size, and only 10 full-size particles are modeled. The number of inclusions is sufficiently increased to achieve a reasonable convergence in the obtained results. The periodic modeling is considered in RVE as shown in the figure. MSC Marc solver was employed to accomplish the numerical simulation. For both phases, linear elastic behavior was considered. Only 1% longitudinal strain was applied to the model to obtain the tensile modulus of the RVE mode. Moreover, the composite material Poisson’s ratio was determined by dividing the transverse to longitudinal strains captured during the FE simulation.

According to the experiment conducted by Srivastava, Maiti [43], 10% weight fraction has been considered as the reinforcement phase. They reported that the tensile modulus of pure PVDF at room temperature and under quasi-static loading condition had been determined 860 MPa. The tensile modulus of CCTO has also been computed 256 GPa according to the experimental measurements conducted by Ramírez, Parra [44]. Hence, the elastic material properties of RVE model consisting of inclusion and matrix are implemented into the FE model with 10% weight fraction.

Since the packing algorithm has a random behavior, five different particle configurations are considered to check the model repeatability. All five samples are filled with 10 particles, and the weight percentages remained 10%. Allowable particle distances are considered in the range of 0.001 to 0.1 related to the particle size. Table 2 gives the determined mechanical properties for all samples. As the table suggests, the tensile modulus is determined 937.85 ± 3.97, which shows only 5.85% error compared with the experimental value (886 MPa) reported by Srivastava, Maiti [43]. It is worth noting that the same density is calculated for all case configurations, proving the model packing validity.

In the present work, Au nanoparticles are dispersed into the PVDF matrix. The mechanical properties of the fabricated nanocomposite are predicted by micromechanical modeling using a RVE cube. The material behavior of Au is assumed as a linear elastic behavior with the tensile modulus of 100 GPa [45], Poisson’s ratio of 0.42 [46], and density of 19,320 kg/m^3^ [47]. Since elastic behavior is studied in this work, only elastic properties are considered for the PVDF phase. All mechanical and physical properties are employed by [43]. According to the test program, 0.05, 0.1, and 0.5% of Au nanoparticles in the PVDF matrix are simulated. As stated before, 0.1% axial strain is applied to the FE model, and tensile and shear moduli and Poisson’s ratio are then numerically predicted.

## 3. Results and Discussions

### 3.1. SEM

SEM was used to study the surface morphology of nanoparticles and nanocomposite films as well. To provide SEM image of nanoparticles, a small amount (a few drops) of prepared colloidal Au nanoparticles in NMP were poured on the glass slide, and the slide was kept in a vacuum oven for 3 h at 120 °C until the NMP was completely removed. Figure 3 shows the SEM image of colloidal Au nanoparticles dried on the glass slide.

The surface morphology of PVDF nanocomposite films was also examined by SEM, as depicted in Figure 4. The SEM image of the PVDF film contains spherulites with about 20 μm in diameter. As it is evident, the spherulitic morphology of pure PVDF is preserved for nanocomposite films. The spherulite size is reduced with an increase in Au nanoparticles content in the PVDF nanocomposite films.

### 3.2. FTIR Spectroscopy

FTIR analysis was performed to investigate the effect of nanoparticle addition on the structure of PVDF polymer. To record the FTIR spectrum, the film sample was pasted on the sample holder in front of the hole so that the light could pass through it. Before each measurement, a background air spectrum was scanned in the same instrumental conditions.

Figure 5 shows the FTIR spectra of the nanocomposite films with different nanofiller contents. The FTIR spectrum of PVDF shows vibration bands at 610, 763, and 975 cm^−1^, used for the α-phase identification. Also, the absorption peak at 431, 840, 1165, 1176 cm^−1^ are the characteristic band of β-phase [48,49,50]. As could be observed in the figure, by increasing the Au nanoparticles concentration, the intensity of the peaks at 610 and 763 cm^−1^ related to α-phase was decreased. On the other hand, the β-phase characteristic peaks at 840 and 1176 cm^−1^ appeared for the sample with the inclusion of 0.5% Au nanoparticles.

The β phase content can be quantitatively determined by using the FTIR spectroscopy according to Equation (1) [15,51]:(1)$$F\left(\beta \right)=\frac{A_{\beta }}{\left(\frac{K_{\beta }}{K_{\alpha }}\right)\;A_{\alpha }+A_{\beta }}$$
where *F(β)* represents the β phase content; *A_α_* and *A_β_* are the absorbances at 763 (α phase) and 840 cm^−1^ (β phase). *K_α_* and *K_β_* are the absorption coefficients at the corresponding wavenumber with the Values of the 6.1 × 10^4^ and 7.7 × 10^4^ cm^2^/mol, respectively [15].

The β phase percentage of PVDF and PVDF/Au nanocomposites are shown in Figure 6. According to the Figure, the PVDF film has the β phase content of 44%. This value is increased by cumulative Au nanoparticles concentration, reaching 54, 59, and 66% for 0.05, 0.1, and 0.1 Au nanoparticle content, respectively. The transformation of α to β phase occurs due to the electrostatic interactions between the CF electric dipoles in the PVDF chain and the surface charge of Au nanoparticles, which can change the polymer chain’s spatial arrangement leading to β-phase formation [31].

### 3.3. XRD

XRD was also performed to study the crystalline structure of PVDF nanocomposite films. XRD patterns of the PVDF nanocomposites films are shown in Figure 7. At XRD pattern of PVDF film, the diffraction peaks at 2θ equal 17°, 18.2°, 19.4°, and 25.2° related to α-phase and assigned to reflections of (100), (110), (020), and (021) planes, respectively [15,31]. The peaks at 2θ equal 20.3° and are associated with β phase describing the reflection of (110)/(200) planes [15,31]. Since the PVDF/0.05% Au spectrum has no perceptible difference compared with PVDF, this sample’s XRD pattern has not been illustrated in the figure.

From the XRD pattern of PVDF/0.1%Au, it was observed that the two peaks at 18.2° and 25.2° completely disappeared and the intensity of the peak located at 17° was significantly reduced. The considerable changes in the crystalline structure of PVDF occurred after the implantation of 0.5% Au nanoparticles into the polymer matrix, so that the peaks at 17°, 18.2° and 25.2° completely vanished, and the diffraction peak at 19.4° moved to 20.3°, which is associated with the crystalline β phase. Generally, the presence of the main peak located at 2θ higher than 20° with no peak at around 25° is clear proof for the domination of β phase inside the polymer matrix [15]. The small peak around 38.5° is ascribed to the (111) plane of Au nanoparticles crystal structure [31].

### 3.4. TGA

Thermogravimetric analysis was conducted to study the thermal stability of the PVDF/Au nanocomposite films. Figure 8 shows TGA thermograms of the PVDF, PVDF/0.5%Au and PVDF/0.1%Au nanocomposites. Since the TGA thermogram of PVDF/0.05%Au is very similar to that of PVDF, it is not shown in Figure 8 to avoid crowding. For the PVDF film, the onset temperature for degradation is found around 409 °C. The onset degradation temperature was shifted to a higher temperature by adding Au nanoparticles and reached 426 and 460 °C for PVDF/0. 5%Au and PVDF/0.1%Au, respectively. These results show that the presence of Au nanoparticles improves the PVDF’s thermal stability, which can be associated with better packing of the crystallites β-phase compared with α-phase. The interaction between PVDF and Au nanoparticles may also result in the improvement of thermal stability.

### 3.5. Dielectric Constants Determination

Dielectric constants of PVDF and PVDF nanocomposites were measured at room temperature in the frequency range of 7.5 × 10^4^ to 2 × 10^6^ Hz, as shown in Figure 9.

As the figure demonstrates, a gradual increment in dielectric constant was observed by adding Au nanoparticles to the PVDF polymer matrix even at such a low concentration. The dielectric constant of PVDF film at 7.5 × 10^4^ Hz is obtained 4.35 increased to 4.6, 5.3, and 5.5 for PVDF/0.05%Au, PVDF/0.1%Au and PVDF/0.5%Au, respectively. As Figure 9 shows, the dielectric constant is achieved proportional to the concentration of Au nanoparticles. In general, the enhancement of dielectric constant may occur for three reasons. First, conductive nanofillers can result in the micro-capacitors network formation in a polymer, leading to a total increase of the dielectric constant of the nanocomposite films compared with pure PVDF [52]. Second, the Maxwell–Wagner–Sillars (MWS) polarization effect, associating with the entrapment of free charges between PVDF and conductive Au nanofiller interface, could increase the dielectric constant [53]. Third, Au nanoparticles cause the β phase in PVDF, resulting in dielectric constant increment due to the larger polarization of the β phase compared to the α phase [54].

### 3.6. Mechanical Properties

Predicted mechanical properties, including tensile modulus, Poisson’s ratio, and shear modulus, are presented in Table 3. By increasing Au nanoparticles content from 0.05 to 0.5%, the material tensile modulus is enhanced by about 0.0091, 0.0181, and 0.0923% compared to the pure PVDF, respectively. It implies that reinforcing PVDF by Au nanoparticles in the range of 0.05 to 0.5% of weight content is slightly improved. The same trend is also obtained for the shear modulus following the well-known relation, G=E/2(1+υ) [55], where G, E and υ are the shear and tensile moduli and Poisson’s ratio. However, the specimen density is enhanced by 0.0455, 0.0910 and 0.4562% for the weight content of 0.05, 0.1 and 0.5% compared with the pure PVDF. The present micromechanical analysis demonstrates that Poisson’s ratio of the fabricated nanocomposite remains unchanged for the considered weight content range.

To make the data more comparable, specific tensile modulus (tensile modulus divided to density) is calculated for 0.05, 0.1 and 0.5% weight content as 0.4830, 0.4828 and 0.4814 MPa·m^3^/kg, respectively. In other word, by increasing the weight content of Au nanoparticles, the specific tensile modulus is reduced by 0.0364, 0.0728, and 0.3622%, respectively, in comparison with the pure PVDF.

## 4. Conclusions

In this research, a simple solution casting method was employed to produce PVDF/Au nanocomposite films with improved crystalline and dielectric properties. The morphology, structural and thermal properties of nanocomposites were studied by SEM, FTIR, XRD, TGA analysis. Remarkable modifications in the polymer’s crystalline structure were detected by adding Au nanoparticles to the polymer matrix. The β-phase volume of the prepared films was taken out from the FTIR spectrum, illustrating the improvement of β-phase fraction with an increase in Au nanoparticles concentration. The presence of Au nanoparticles alters the thermal constancy of the polymer matrix. By adding Au nanoparticles, the temperature of onset degradation increases from 409 °C for the pure PVDF to around 460 °C for PVDF/0.5%Au nanocomposite film. The dielectric constant of nanocomposite films, measured by LCR meter, was increased by increasing nanoparticle concentration. Furthermore, the micromechanical model showed that the specific tensile modulus is reduced by 0.0364, 0.0728, and 0.3622% by increasing the Au nanoparticles’ weight content of 0.05, 0.1, and 0.5%, respectively.

## Figures and Tables

**Figure 1 polymers-12-02630-f001:**
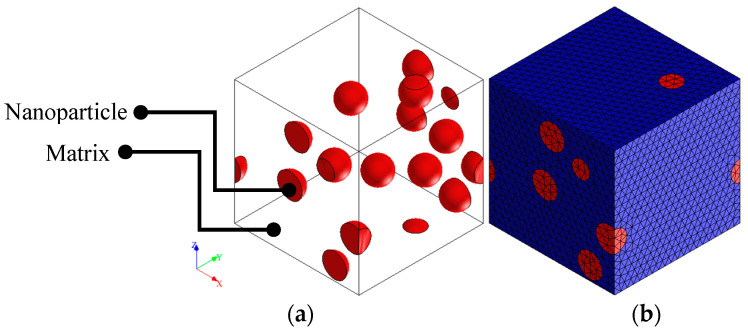
Micromechanical model of nanoparticle inclusion in the PVDF matrix (**a**) 3D model, (**b**) FE constructed mesh.

**Figure 2 polymers-12-02630-f002:**
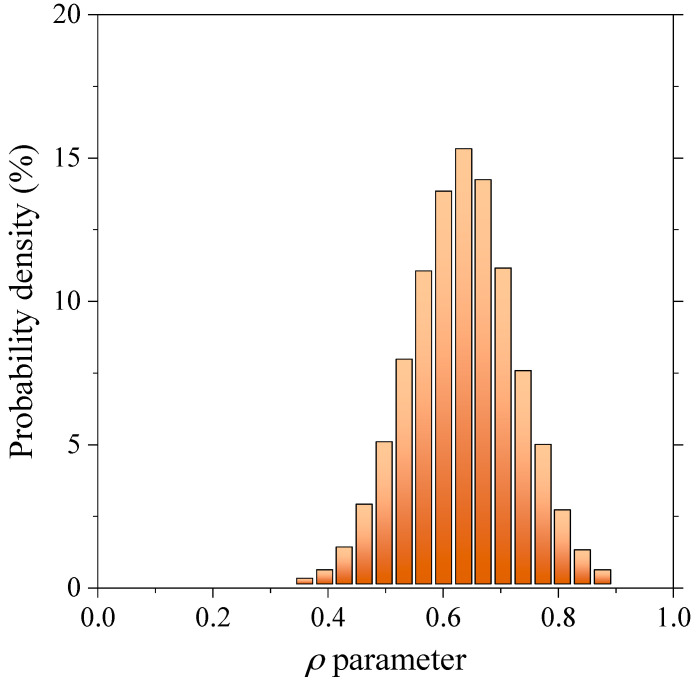
Normal distribution of ρ parameter of FE constructed model.

**Figure 3 polymers-12-02630-f003:**
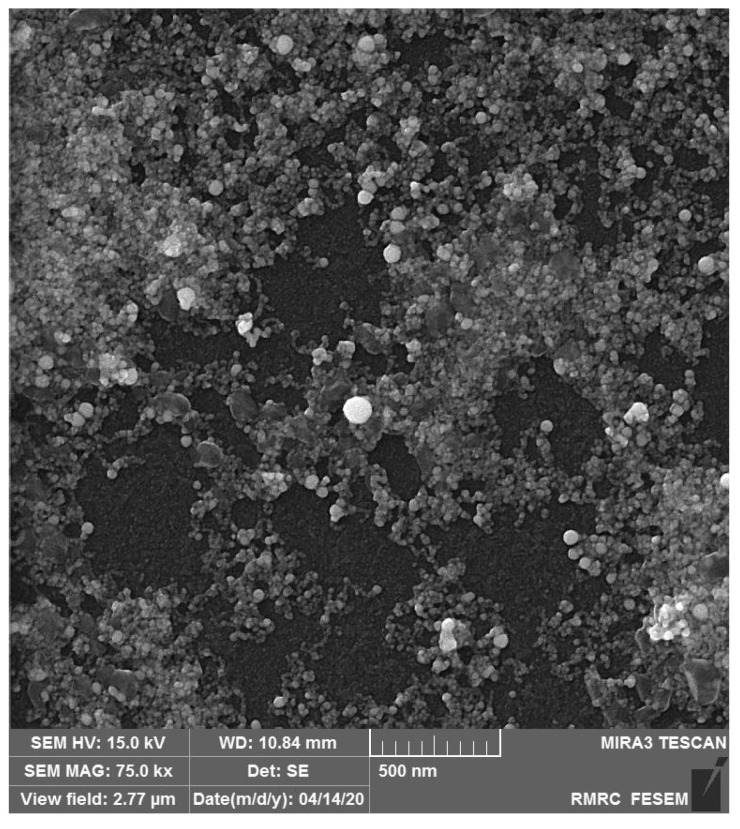
SEM image of Au nanoparticles.

**Figure 4 polymers-12-02630-f004:**
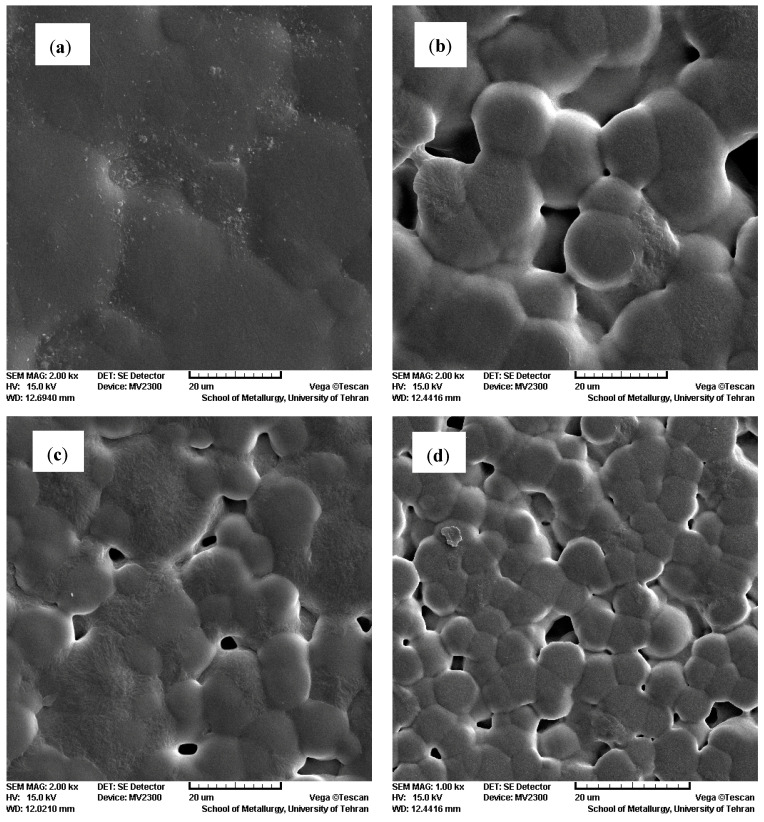
The SEM images of (**a**) PVDF, (**b**) PVDF/0.05%Au, (**c**) PVDF/0.1%Au and (**d**) PVDF/0.5%Au.

**Figure 5 polymers-12-02630-f005:**
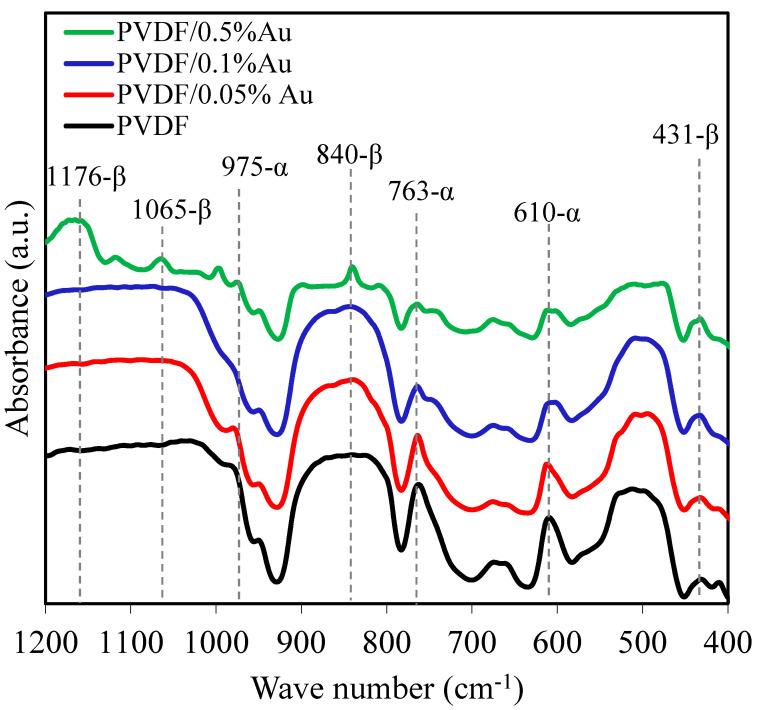
FTIR spectra of PVDF and PVDF/Au nanocomposite films.

**Figure 6 polymers-12-02630-f006:**
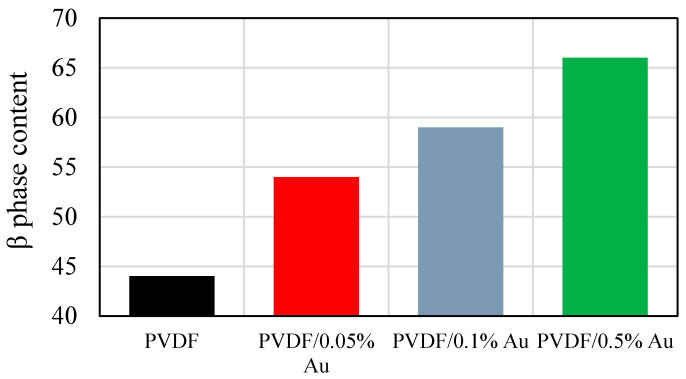
The β phase content of PVDF and PVDF/Au nanocomposite films.

**Figure 7 polymers-12-02630-f007:**
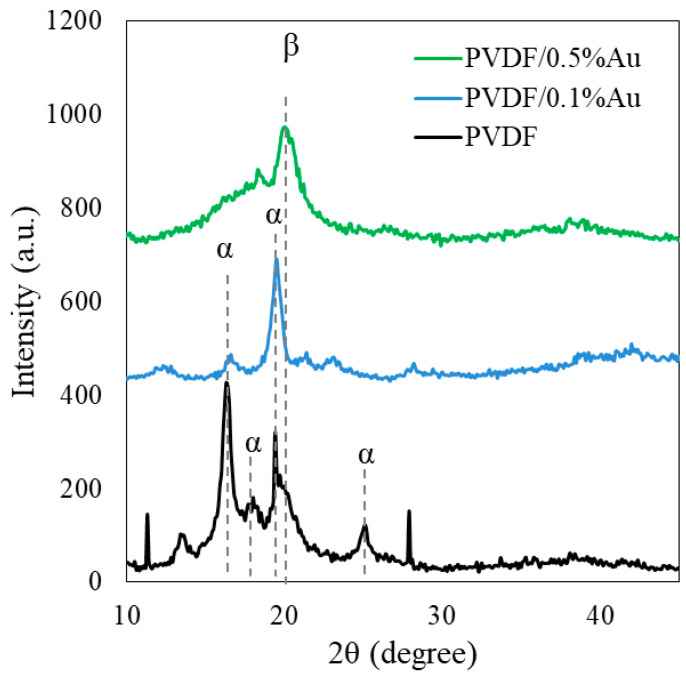
XRD pattern of PVDF and PVDF/Au nanocomposites.

**Figure 8 polymers-12-02630-f008:**
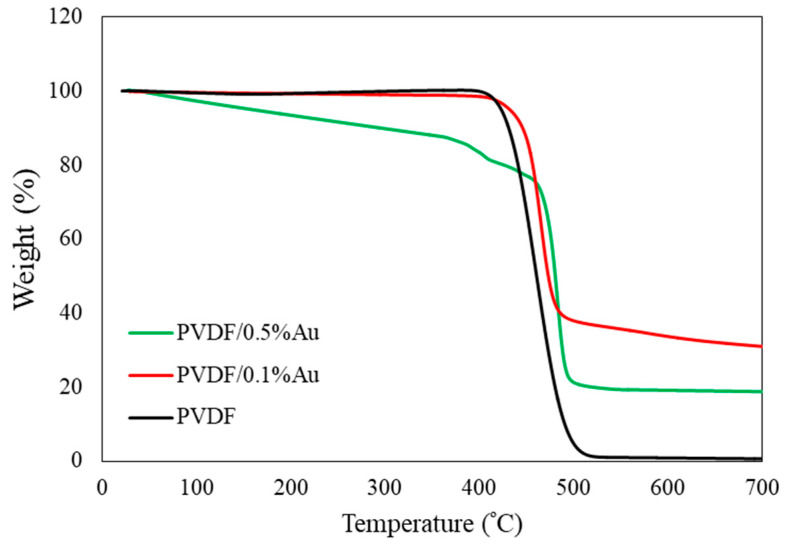
TGA thermograms of PVDF and PVDF/Au nanocomposites films.

**Figure 9 polymers-12-02630-f009:**
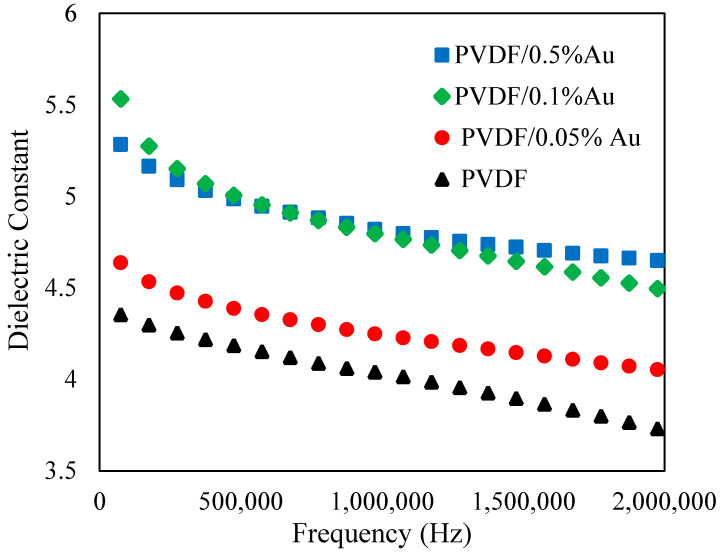
Dielectric constant PVDF and PVDF/Au nanocomposites.

**Table 1 polymers-12-02630-t001:** Nanocomposites based on PVDF and Au nanoparticles from previous literature.

Year	Composite	Method for the Synthesis of Au Nanoparticles	Focus	Ref.
2011	PVDF + Au NPsPVDF + Au NS	Reduction of HAuCl_4_	Polymorphism change	[28]
2012	PVDF-Au	Reduction of HAuCl_4_	Polymorphism change + Thermal FTIR study	[29]
2016	Au NP-MWCNT-PVDF	Reduction of HAuCl_4_	Electromagnetic Interference Shielding	[30]
2016	PVDF-GO/Au NPs	Reduction of HAuCl_4_	Polymorphism change + β-phase content + dielectric properties	[31]
2017	Au-PVDF(pp)	-	Dielectric properties	[32]
2017	Au NPs/PVDF	Laser ablation	Enhancement of polar phase	[33]
2019	ES Au-PVDF	Reduction of HAuCl_4_	The voltage and Current output	[34]
2019	PVDF-Au NPs	Reduction of HAuCl_4_	β-phase polarization behavior	[35]
2019	Au-BaTiO_3_/PVDF	Reduction of HAuCl_4_	Dielectric properties	[36]

**Table 2 polymers-12-02630-t002:** Variation of mechanical properties obtained by micromechanical analysis for different particle configurations.

Case Number	1	2	3	4	5	Average
Weight fraction (%)	10	10	10	10	10	10
Number of inclusion	10	10	10	10	10	10
Allowable relative particle distance	0.001	0.003	0.005	0.01	0.1	0.0238 ± 0.04
3D model of RVE	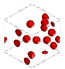	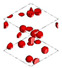	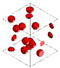	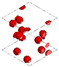	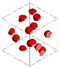	
Tensile modulus (MPa)	938.45	935.75	933.57	945.17	936.33	937.85 ± 3.97
Density (kg/m^3^)	1899.78	1899.78	1899.78	1899.78	1899.78	1899.78

**Table 3 polymers-12-02630-t003:** Predicted mechanical properties of PVDF-Au nanoparticles nanocomposite at different weight content.

Weight Fraction of Au Nanoparticles (%)	0.05	0.1	0.5
Tensile modulus (MPa)	860.08	860.16	860.80
Shear modulus (MPa)	320.93	320.96	321.21
Poisson’s ratio	0.34	0.34	0.34
Density (kg/m^3^)	1780.81	1781.62	1788.12

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
