# Peer review of "Laser Ablation-Assisted Synthesis of Poly (Vinylidene Fluoride)/Au Nanocomposites: Crystalline Phase and Micromechanical Finite Element Analysis"

_polymers, 2020, doi:10.3390/polym12112630_

Round 1

Reviewer 1 Report

The manuscript describes how the properties of PVDF with respect to crystallinity and micromechanical properties may be improved due to the addition of gold nanoparticles. With respect to piezoelectric applications the fraction of the b crystalline phase is important. It is discussed that the presence of gold nanoparticles is significantly enhanced. The morphology of the composites was investigated by SEM and thermal stability by TGA. Micromechanical properties were derived from finite element analysis.

Despite the fact that the topic of the manuscript is of high importance the manuscript has quite a number of weak points. Thus, I recommend to reconsider the manuscript after major revision.

Critical points are the following:

Experimental details are not given in full detail. For example, sample preparation for FTIR and SEM is not explained well. With respect to SEM it is mentioned that the film was put on a lamella. More information is required. With respect to FTIR the configuration of the instrument is missing, e.g., source, detector, and resolution. In addition, it is not stated how the spectra were recorded. The film was measured through plane, mounted on an ATR crystal? This information must be provided.

With respect to the FTIR results:

The spectra are given in Figure 4. Firstly, the spectrum for PVDF containing 0.5 % Au looks really strange. What is the meaning of the many little peaks observed for wavenumbers higher than approximately 950 cm−1? Moreover, it is stated that the b phase content calculated from the peak at 840 cm−1 is increasing. However, I cannot see any peak at 840 cm−1 in the case of no, 0.05, and 0.1 % Au. For the highest Au content there is no clear peak, too.

The authors have to explain how they determined the absorbances introduced into equation (1). Did you deconvolute the spectra? Did you subtract a baseline or how did you analyze the spectra? It is not convincing that the sample without Au has more than 40 % of b phase according to Figure 5, however, there is no peak in the spectrum in Figure 4.

In short: I am not convinced by the analyses of the FTIR data. This part requires significant rewording and better explanation.

With respect to the XRD data in Figure 6 and the associated text:

The peaks in the spectra have to be assigned clearly to the various crystalline phases. The important peak assigned to the b phase as well as the peaks associated with the alpha phase have to given. The paragraph below Figure 6 reads “The considerable changes in the crystalline structure of PVDF occurred after the implantation of 0.5 % Au.” Figure 6 shows that already 0.1 % of Au leads to significant changes.

The data indicate the transformation to mainly b phase PVDF. However, the description in the text must be improved.

With respect to TGA:

In the text it is stated that the data for PVDF/0.5% Au is not contained in the plot. According to the plot the 0.1 % Au sample is missing – if the legend is correct.

Lines 241-242: “the dielectric constant of PVDF film at 7.5x104 Hz is obtained 4.35 increased to 4.6….” The sentence requires rephrasing an I assume it must read 10 to the power of 4.

Lines 259 to 261: “However, the specimen density is enhanced by 0.0455, 0.0910 and 0.4562 for weight content of ….” I think units are missing. “micromechanical analysis is shown that” requires rephrasing.

Table 2: The number of significant digits in the table is between 5 and 8. I doubt that these numbers are significant. The numbers should be reduced to a physical meaningful number of significant digits.

The Abstract requires some rephrasing. For example “has been occurred”, “the polar phase fraction was also significantly improved”, “to have extended applications for practical usage”. The latter is actually having no content.

In general, the English language is quite variable throughout the manuscript and needs to be homogenized.

Small points:

The wording of the title appears a bit strange to me: Crystalline properties Improvement….

Line 79: the units of the molar mass must be given

Line 80: the company Merck is misspelled.

Line 33: correct to vinylidene

Line 13: correct to poly(vinylidene fluoride)

in general: N-Methyl-2-…: methyl should be spelled with a non-capital m

Finally, I cannot judge the finite element simulation and its results, because this is certainly not my expertise. The presentation appears to be fine to me as a non-expert.

Reviewer 2 Report

Micromechanical Finite Element Method analysis description should be definitely described in details (finite element types, total number, degrees of freedom, meshing algorithm, solution procedure as well as overall computer power engaged to solve the problem). 

The overall lookout of the article should be more scientific, Fig. 9 is more plausible for advertisement than for the research paper. 

Similar computer simulations have been presented before, see at least in SokoÅ‚owski D., KamiÅ„ski M., Homogenization of carbon/polymer composites with anisotropic distribution of particles and stochastic interface defects. Acta Mech. 229(9): 3727-3765, 2018. 

Reviewer 3 Report

This work doesn't have any novelty and the way of presenting data is not appropriate for research work and publication.

Round 2

Reviewer 2 Report

The Authors have improved their manuscript sufficiently and now it is almost ready for publication. Please, look over any editing discrepancies.

Reviewer 3 Report

The paper now is ready to be published.